



# 1 Permafrost degradation of peatlands in northern Sweden

**Authors**: Samuel Valman[1,2]*, Matthias Siewert[3]*, Doreen Boyd[2], Martha Ledger[4,5], David
Gee[6], Betsabe de la Barreda-Bautista[4,2], Andrew Sowter[6], Sofie Sjogersten[4]
**Affiliations**:
[1] Nottingham Geospatial Institute, University of Nottingham, Nottingham NG7 2TU, UK
[2] School of Geography, University of Nottingham, University Park, NG7 2RD, Nottingham, UK
[3] Department of Ecology and Environmental Sciences, Umeå University, Umeå, Sweden
[4] School of Biosciences, University of Nottingham, Sutton Bonington Campus, College Road, LE12 5RD,
Loughborough, UK
[5] School of Biological Sciences, Kadoorie Biological Sciences Building, University of Hong Kong, Pok Fu Lam
Road, Hong Kong
[6] TerraMotion, Ingenuity Centre, Triumph Rd, Nottingham NG7 2TU
*These authors contributed equally to this work.
Corresponding author: Sofie Sjögersten sofie.sjogersten@nottingham.ac.uk
**Abstract**. Climate warming is degrading palsa peatlands across the circumpolar permafrost region. Permafrost
degradation may lead to ecosystem collapse and potentially strong climate feedbacks, as this ecosystem is an
important carbon store and can transition to being a strong methane emitter. Landscape level measurement of
permafrost degradation is needed to monitor this impact of warming. Surface subsidence is a useful metric of
change and can be monitored using InSAR satellite technology. We combined InSAR data, processed using the
ASPIS algorithm to monitor ground motion between 2017 and 2021, with optical and LiDAR data to investigate
the rate of subsidence across palsa peatlands in northern Sweden. We show that 55% of the area of Sweden's
eight largest palsa peatlands is currently subsiding, which can be attributed to these permafrost landforms and
their degradation. The most rapid degradation occurring in the largest palsa complexes in the most northern part
of the region of study, also corresponding to the areas with the highest % palsa cover within the overall mapped
wetland area. Further, higher degradation rates were found in areas where winter precipitation has increased
substantially. The roughness index calculated from a LiDAR-derived DEM, used as a proxy for degradation,
increases alongside subsidence rates and may be used as a complementary proxy for palsa degradation. We
show that combining datasets captured using remote sensing enables regional-scale estimation of ongoing
permafrost degradation, an important step to-wards estimating the future impact of climate change on
permafrost-dependent ecosystems.

Keywords: Permafrost, subsidence, Arctic, InSAR, palsa, peatlands




## 1.0    Introduction

Permafrost regions are critical components in the climate system, due to their essential carbon (C) storage
service (Harris et al., 2022). The circumpolar permafrost region in particular, stores around 1300±200 Pg of
organic C, corresponding to around 50% of the global terrestrial C pool (Hugelius et al., 2020; Köchy et al.,
2015). It covers around 21 million km² or 22% of the Northern Hemisphere's landscapes (Obu, 2021). Northern
peatlands themselves store an estimated 415±150 Pg of C in an area covering around 3.7 million km² of which
around 1.7 million km² is permafrost substantially overlapping with the circumpolar permafrost region
(Hugelius et al., 2020). Permafrost in these peatlands raises the surface above the water table forming so-called
palsa (pl. palsas) or, in extended form peat plateaus (Seppälä, 2011). These account for substantial areas of
global permafrost, including in northern Fennoscandia (Ballantyne C. K., 2018; Gisnås et al., 2017; Tarnocai et
al., 2009), for example, in northern Sweden 137 km² of this palsa peatland has been reported (Backe, 2014).
Climate warming, and the associated alteration in the precipitation regime, is increasingly recognized to be a
particular threat to permafrost (Biskaborn et al., 2019), with the subarctic Fennoscandian permafrost region, and
the palsa peatlands within, particularly vulnerable (Christiansen et al., 2010; Farbrot et al., 2013).
Climatic models project unsuitable conditions for permafrost within the coming century, with the most pessimistic
estimates projecting unsuitability even sooner - by 2040 (Chadburn et al., 2017; Fewster et al., 2022; Könönen et
al., 2022; Stefan et al., 2006). As palsa peatlands are often found in the sporadic or discontinuous permafrost zone
(Zuidhoff & Kolstrup, 2000), they are particularly sensitive to climate warming and any resultant permafrost thaw
and disappearance. Their sensitivity mainly results from the alterations in the thermal insulation effect of peat
deposits and snow as the climate changes (Seppälä, 2011; Smith & Riseborough, 1996). Specifically, organic peat
has a high thermal conductivity when wet and frozen, but low conductivity when dry and thawed. Snow has a
highly insulating effect on ground temperature. Thus, extended periods of air-temperatures below 0°C and thin
snow cover in winter are beneficial to maintain or grow the frozen permafrost core of palsas and peat plateaus.
Low summer precipitation, which reduces the thermal conductivity of peat, also helps to preserve the frozen cores
in palsa. In contrast, increased snowfall has been linked to permafrost degradation as it increases winter insulation.
Further, high summer precipitation leads to higher thermal conductivity of peat, and combined with warm summer
temperatures, can degrade permafrost by increasing permafrost temperatures and subsequent thawing of the frozen
peat core of palsas. The strong insulating properties of peat allow the occurrence of permafrost at the southern
extent of the northern permafrost region and valley bottoms in areas otherwise too warm for permafrost (Johansson
et al., 2013; Seppälä, 2011; Smith & Riseborough, 1996).
Warming of the permafrost in palsa peatlands typically leads to top-down thaw, (i.e. thickening of the active
layer), and eventual subsidence of the surface, as well as lateral thaw, sometimes called abrupt thaw or
thermokarst, which occurs at the margin of peat plateaus (Seppälä, 2011; Smith & Riseborough, 1996; Zuidhoff,
2002). This is often associated with water-logged conditions and, as a result, increased methane (CH₄) emissions
(Glagolev et al., 2011; Hugelius et al., 2020; Matthews et al., 1997; Miglovets et al., 2021; Schuur et al., 2009;
Turetsky et al., 2020; Varner et al., 2022), which is a central theme for permafrost research (Sjöberg *et al.*,
2020). A subsequent impact of this permafrost degradation is an alteration in vegetation cover, its hydrology,
and human use of the landscape (e.g., infrastructure and reindeer husbandry)(Markkula et al., 2019; Ramage et
al., 2021). Given the potentially large impacts of permafrost thaw on the global climate, ecosystem function and
human activity, quantification and monitoring of the subsidence in peat deposits affected by permafrost thaw
and degradation, as well as an understanding of their sensitivity to changing climatic parameters, is urgently
required (IPCC, 2021).
The degradation of the permafrost of palsa peatlands has been observed right across the circumpolar permafrost
region in a number of studies, including in northern Scandinavia (Åkerman & Johansson, 2008; de la Barreda-
Bautista et al., 2022; Luoto & Seppälä, 2003; Olvmo et al., 2020; Sannel et al., 2016; Varner et al., 2022);
Russia (Glagolev et al., 2011; Miglovets et al., 2021; van Huissteden et al., 2021); the USA (Douglas et al.,
2021; Douglas et al., 2015; Sannel, 2020) and Canada (Mamet et al., 2017; Sannel & Kuhry, 2011; Short et al.,



2014; Vallée & Payette, 2007). Although rapid degradation in response to short term climatic events has been
observed, typically permafrost degradation has been investigated via long-term monitoring at decadal timescales
in response to changes in temperature and precipitation conditions (Åkerman & Johansson, 2008; de la Barreda-
Bautista et al., 2022; Olvmo et al., 2020; Sannel et al., 2016). These longer-term studies have shown strong
relationships between permafrost degradation and summer temperatures, length of the thaw period, winter
precipitation and snow depth (Smith et al., 2022). These types of analyses are very useful for quantifying how
much of the landscape has already transitioned and understanding the climate change drivers behind these
changes, but they do not capture the initial stages of permafrost degradation in palsa peatlands and the lower
rates of subsidence that have yet to result in observable changes in the vegetation or thermokarst formation. The
latter is crucial to understand the ongoing response of palsa peatlands to climate warming and to predict when
pulses of greenhouse gases to the atmosphere and other impacts (e.g., on infrastructure) are likely to occur.
Thus, approaches that detect early signs of degradation at landscape scales, with repeated observations, are
urgently required.
Due to the vast extent and remoteness of permafrost areas, we looked to satellite remote sensing to underpin the
measurement and monitoring assessment of permafrost peatlands, their degradation and resultant climate
impacts (Armstrong McKay et al., 2022; Hugelius et al., 2020; Obu, 2021; Schuur et al., 2015; Swingedouw et
al., 2020). Optical remote sensing approaches can be augmented with RaDAR remote sensing methods,
including InSAR, to capture the early response of permafrost to warming, since these methods can detect
vertical land surface motion at millimetre precision across natural landscapes (Alshammari et al., 2020;
Alshammari et al., 2018; de la Barreda-Bautista et al., 2022; Short et al., 2014; van Huissteden et al., 2021)
(Bartsch *et al.*, 2016). The regular sampling frequency, insensitivity to cloud and, in the case of Sentinel-1, low
cost, means InSAR from Sentinel-1 should be well suited to measure and monitor ongoing changes in
permafrost affected by climate change. Further, Sentinel-1 for InSAR is effective at both local and regional
scales - the 20m × 20m spatial resolution enables measurement of surface motion within local sites (de la
Barreda-Bautista et al., 2022), and can do so over entire and complex landscapes, such as the circumpolar
permafrost region (Reinosch et al., 2020).
The overall aim of this study was to carry out a regional-scale analysis of permafrost degradation across the
palsa peatlands of northern Sweden, principally using Sentinel-1 InSAR-derived subsidence as an indication of
degradation. Pertinent to this is that any InSAR-detected changes can be associated with known and delineated
targets in the wider landscape. Furthermore, it is also important to understand any within-site dynamics of
permafrost degradation. This paper therefore has specific objectives to: (i) measure the subsidence rate between
2017-2021 of all major palsa peatlands in the northern Sweden region; (ii) determine in which palsa peatlands
subsidence is greatest, and (iii) assess if the spatial patterns of degradation can be linked to climatic variables
and properties of the different sites across the region. To achieve these objectives, we combined large-scale
regional analysis with higher resolution site-specific analysis of patterns in subsidence, using a combination of
datasets - satellite (Sentinel-1) InSAR; occupied airborne optical and LiDAR; and snow depth, precipitation, and
temperature time-series from meteorological stations across the region.

**2.0 Methodology**
**2.1 Study area**
This study focused on the northern part of Sweden; a region containing palsa peatlands, located between 68.84-
67.64° N and 18.71-21.19° E. The palsa peatlands of the region are confined predominantly to valley bottoms in
an elevation range between ca. 350 and 590m asl (Fig. 1). The rest of the study area region is comprised of
forests and/or mountain land covers (Siewert, 2018). Of all the palsas in the region, the eight largest palsa
peatlands complexes range between 50 and 273ha in area (Table 1). These were located across the region, which
covers a ca. 20,000km$^2$ area, with the largest palsa sites located in the north-western parts of the region. Smaller
palsa peatlands occur scattered in distribution right across the region. The climate varies across the region from
north to south ([www.smhi.se](www.smhi.se)). The mean January and July temperatures in Karesuando in the northern part of



the study region is -16 and 12.8°C, respectively, while in Kiruna, slightly further south, the mean January and
July temperatures is -11.6 and 13.4C (1991-2020 average). Mean annual precipitation is 443 and 560mm in
Karesuando and Kiruna, respectively.

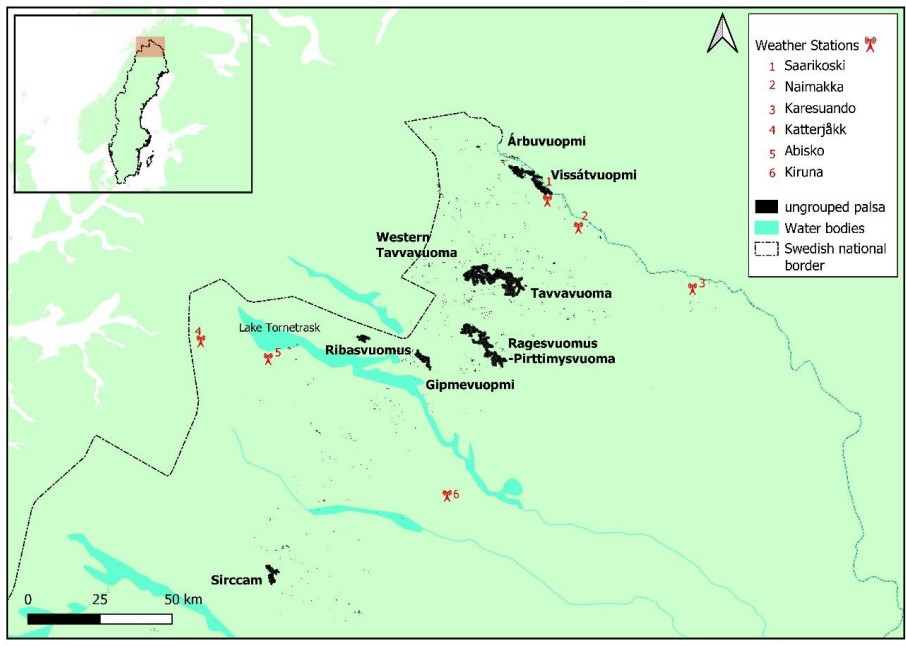


*Figure. 1: Map of the palsa peatland complexes in Sweden which were investigated in the study focusing on the
eight named palsa peatlands. The black regions show where 250m buffers around the palsa areas have created
continuous expanses (Backe, 2014). Meteorological station positions used in the study are also indicated.*


We selected larger palsa areas of the region to focus our analysis. This was in line with focus areas by the mapping
of palsas undertaken as part of a previous national palsa peatland mapping effort (Backe, 2014). The resultant
palsa peatland mapping dataset has a spatial resolution of 100m, with the % palsa cover for each pixel computed,
and these pixels given a 250m buffer to produce continuous area outputs. The eight largest continuous areas of
these palsa peatlands from the national palsa mapping dataset were selected for this study (Backe, 2014), hereon
in referred to as palsa complexes, a term reflecting their mosaic nature of raised palsa plateaux, interspersed with
lower lying fen or thermokarst areas. This afforded analyses at a spatial resolution suitable for analysis with
Sentinel-1 yet provide practical representation of the condition of the peatland in the region. These eight sites
account for the majority of the palsa peatland areas in Sweden, the sites are listed in Table 1 along with some
associated information on their status and total and raised palsa plateau areas.

*Table. 1: Information on the major palsa complexes analysed in this paper (Backe, 2014). The protection status
means no or limited direct anthropogenic activities that may influence palsa degradation. Total site area is
calculated from the total number of 100m × 100m palsa pixels at each site - these pixels have associated
percentages for how much of the 100m x 100m area is palsa. The average of these percentages for each site
displays the palsa density at each site. These percentages are then used to calculate the "total palsa area" for
each site based on the original report estimates.*



| Site Name | Protection Classification | Total site area (ha) | Average extent palsa in these areas (%) | Total palsa area (ha) | Central location (Latitude, Longitude) |
|---|---|---|---|---|---|
| Árbuvuopmi | Not protected | 327 | 26.3 | 86.06 | 21.03464, 68.83842 |
| Vissátvuopmi | Not protected | 867 | 31.6 | 273.75 | 21.19497, 68.79412 |
| Tavvavuoma | EU Nature 2000 SPA, SAC. Site of National Importance for Nature conservation | 1719 | 15.8 | 271.25 | 20.85043, 68.51132 |
| Western Tavvavuoma | EU Nature 2000 SPA, SAC. Site of National Importance for Nature conservation | 813 | 13.0 | 105.74 | 20.57727, 68.53953 |
| Gipmevuopmi | Pristine mountain forest, Nature reserve, EU Nature 2000 SCI | 303 | 23.0 | 69.62 | 20.09767, 68.28377 |
| Ragesvuomus-Pirttimysvuoma | Pristine mountain forest, Nature reserve, EU Nature 2000 SCI | 881 | 6.55 | 57.74 | 20.48660, 68.3741 |
| Sirccam | EU Nature 2000 SCI | 397 | 12.8 | 50.70 | 18.71528, 67.64537 |
| Ribasvuomus | Pristine mountain forest, Nature reserve, EU Nature 2000 SCI | 216 | 23.2 | 50.13 | 19.60100, 68.36116 |



## 2.2 Datasets

The InSAR-derived dataset of surface motion over this northern Sweden region of study was calculated for the period between 2017 to 2021, from single look complex C-band SAR data, captured in Interferometric Wide Swath mode by the Sentinel-1 constellation (European Union's Copernicus Programme; Torres et al., 2012). SAR data input were from the thaw season when there was minimal coverage of snow and ice (i.e., between April and October in each year). Data from descending tracks 168 and 66 were used to cover the target area. Four stacks were processed independently with one from track 168 and three from track 66, which was split into a northern, middle, and southern subsets. The APSIS (formerly ISBAS) method (Sowter et al., 2013; Sowter et al., 2016) was used to characterize surface motion which relaxes the need for consistent phase stability and therefore enables near-complete spatial and temporal coverage over vegetated surfaces (Alshammari et al., 2020; Alshammari et al., 2018; Bradley et al., 2022; Cigna & Sowter, 2017; Gee et al., 2017; Sowter et al., 2016), including those found across snow-free permafrost regions.

InSAR processing involved the co-registration of each Sentinel-1 image to a common slant range coordinate system and multi-looking of data by factors of 7 in range and 2 azimuth. This produced a dataset with an approximate spatial resolution of 20m × 20m. Using a perpendicular baseline of 250m and maximum temporal baseline of 183 days ∼ 2100 interferograms were generated per stack. The temporal baseline was chosen to balance the need to reduce the baseline to minimise phase ambiguities and best maintain coherence across the region, whilst also using a baseline long enough to generate season-to-season pairs over consecutive years. This is required over permafrost regions to capture more subtle trends of surface motion during the thaw period (de la Barreda-Bautista et al., 2022; Liu et al., 2010). The interferograms were unwrapped using a modified version of the SNAPHU algorithm. The multi-annual average velocity was calculated for pixels which maintained a coherence greater than 0.45 in a minimum of ∼ 650 interferograms, with respect to stable reference points located in the town Kautekenio (N°69.00, E°23.04) for track 168 and Narvik (N°68.44, E°17.42), Kvikkjokk (N°66.95, E°17.72), and Rognan (N°67.09) for the subsets of track 66. The line-of-sight measurements were converted to vertical surface displacement using a cosine correction and finally mosaicked into a single deformation product. Localised UAV studies at sites in Sweden have verified the accuracy of using InSAR to monitor permafrost degradation (de la Barreda-Bautista et al., 2022).



In order to interpret the resultant surface motion dataset produced by the ASPIS InSAR method, two sets of
additional data were sourced: (i) higher resolution remote sensing data and (ii) meteorological data. The former
included orthophotos captured of the eight target areas by occupied airborne surveys commissioned by the
Swedish Survey (www.lantmateriet.se; © Lantmäteriet). The orthophotos were panchromatic, with each scene
covering a 5km × 5km area, at a 0.5m spatial resolution, the majority were captured in 2016, although gaps were
filled with imagery from 2010 and 2008. The Swedish National Digital Elevation Model (DEM), was also used
in this study. The DEM was derived via occupied airborne LiDAR data capture in 2016 and processed to compute
elevation at 2m spatial resolution across Sweden (www.lantmateriet.se; © Lantmäteriet). The orthophotos and
DEM provided elevation and landscape characteristics (geomorphic features) for use in this study. The
meteorological data was captured by the Swedish Metrological and Hydrological Institute (www.smhi.se) at
meteorological stations across the region. Specifically, the air temperature, precipitation, and snow depth data,
were sourced and used from specific stations, i.e., those located closest to the palsa complexes under investigation
namely at Katterjåkk, Abisko, Kiruna and Karesuando, Saarikoski/Naimakka (Fig. 1).

## 2.3 Data analyses

The ASPIS InSAR surface motion dataset was resampled using the mean value from the original 20m × 20m to
match the 100m × 100m spatial resolution of the palsa peatland dataset which makes up the eight palsa complexes
(Backe 2014). From this the frequency distributions of ASPIS InSAR surface motion at these eight palsa
complexes, and over all individual palsa peatland pixels in the region, were produced. Using these data, the
maximum and minimum rates of surface motion at each site was determined, as well as the sum of the pixels with
palsas that showed subsidence. These derived data relating to surface motion were further interpreted using the
orthophotos and DEMs, supported by the meteorological data.
The DEM tiles were joined together and clipped to the eight palsa complexes. Following this, the degree of
elevation roughness was calculated, via the native topographic roughness index function (Riley, DeGloria, &
Elliot, 1999). This roughness index was thresholded at > 0.5 to provide a visual depiction of palsa landform edges
in the otherwise typically even terrain of the valley bottoms where the palsas occur. The roughness data was
visually compared to the orthophotos from a subset of areas to assess its potential for delineating palsas and this
allowed us to determine a threshold value that connected these continuous terrain variables to the specific features
of the palsa complexes, such as the raised mound structure of the palsa – so-called palsa mounds (Franklin, 2020).
Hillshade was also calculated via the native QGIS function using the default formula, which uses a lighting effect
to visualise the roughness of the terrain from differences in local elevation (QGIS, 2022). The roughness,
hillshade, and elevation outputs were overlaid on the mapped palsa tiles to provide higher resolution visual
interpretation. The roughness and elevation outputs were also resampled to the resolution of the mapped palsa
tiles (100m x 100m) to enable statistical comparison. The zonal statistics tool was used to extract mean average
values from the resulting roughness and elevation outputs for the 100m spatial resolution mapped palsa tiles.
Mean annual, maximum, and minimum daily air temperature, precipitation, and depth of ground snow for the
period 2000 to 2022 from the meteorological station nearest to a correspondent palsa complex were extracted and
analysed. The Naimakka station did not provide snow depth and the Saarikoski station did not provide air
temperature, however, it was deemed that at the regional scale of this study these sites were sufficiently close
together (18km) to be interchangable. Subsequently, data was averaged to provide an annual measurement of each
meteorological variable for each station/palsa complex. Due to incomplete meterological datasets, a longer-term
record of the meteorological variables was not possible for all sites. However, long-term climate data (>100years)
was available from three meteorological stations in the region: namely, Karesuando, Kiruna, and Abisko. This
data was used to assess temporal variability in annual, winter (December, January and February (DJF)) and
summer (June, July, and August (JJA)) temperature, precipitation and snowfall since the start of records across
the region. Descriptive statistics (mean, minimum, maximum and inter-quartile range) were produced to express
the regional differences between these sites. Lastly, to complement the point based meteorological (both weather
and climate) data, we used modelled permafrost probabilities based on climatic conditions to explore relationships
between climatic conditions and subsidence rates (Obu et al., 2018). In this context, it is worth noting that there
may be a mismatch between the modelled permafrost distribution and permafrost in palsa peatlands as this can,
in some areas, be a relic of cooler climatic conditions. The mean values from these data on permafrost probability
were used to resampleto a 100m spatial resolution to enable comparison with the other data sets.





To analyse the relationships between surface motion, roughness and percent palsa in each 100m by 100m pixel
stratified by palsa complex, SciPy statistics (Virtanen et al., 2020) was used to obtain Pearson's correlation
statistics. Pandas (McKinney, 2011) and NumPy (Harris et al., 2020) were used for data management. All scripts
are available on the project GitHub (https://github.com/SamValman/Permafrost_Sweden). The relationship
between the meteorological variables both over the last two decades at the weather stations closest to the palsa
complexes and duration of the climate record at the three weather stations with the longest data series were
assessed using regression analysis in Genstat (VNS Ldt). Some of time series were incomplete, in these instances
the analysis was conducted using the slightly shorter time series.

**3.0 Results**
The ASPIS InSAR-derived surface motion outputs for the time-period of interest (2017-2021), ranged between -
9.9 and 7.7mm yr$^{-1}$ across all of the palsa measured, with a mean of 0.05, median of 0.2 and range of 17.7mm yr$^{-1}$
. Focusing solely on the eight palsa complexes provided greater insight and excluded the most extreme uplift
values from scattered individual palsa (Table 2).

*Table. 2: InSAR subsidence and uplift measurements of the palsa complexes defined in Figure 1 and Table 1.*
*The total palsa area were used to isolate and extract ASPIS InSAR measurements of surface motion at each of*
*the eight sites.*

| Site | Max subsidence (mm yr$^{-1}$) | Max uplift (mm yr$^{-1}$) | Subsiding area (ha) | Area subsiding >3.5 mm yr$^{-1}$ (ha) |
|---|---|---|---|---|
| Árbuvuopmi | -9.9 | 1.7 | 321.3 | 138.4 |
| Vissátvuopmi | -8.9 | 3.5 | 796.2 | 204.8 |
| Tavvavuoma Western | -6.4 | 6.6 | 1009.4 | 50.9 |
| Tavvavuoma | -5.1 | 6.3 | 215.0 | 1.0 |
| Gipmevuopmi | -6.9 | 6.3 | 117.2 | 1.8 |
| Ragesvuomus-Pirttimysvuoma | -5.9 | 5.7 | 358.6 | 7.4 |
| Sirccam | -3.1 | 5.4 | 135.3 | 0.0 |
| Ribasvuomus | -6.5 | 5.5 | 93.6 | 0.7 |



The spatial plots of surface motion for each palsa complex displayed in Figure 2, illustrates a spatiality in terms
of surface motion (both subsidence and uplift and associated variance) across this northern Sweden region. This
is evident both within the palsa complexes and between the complexes.



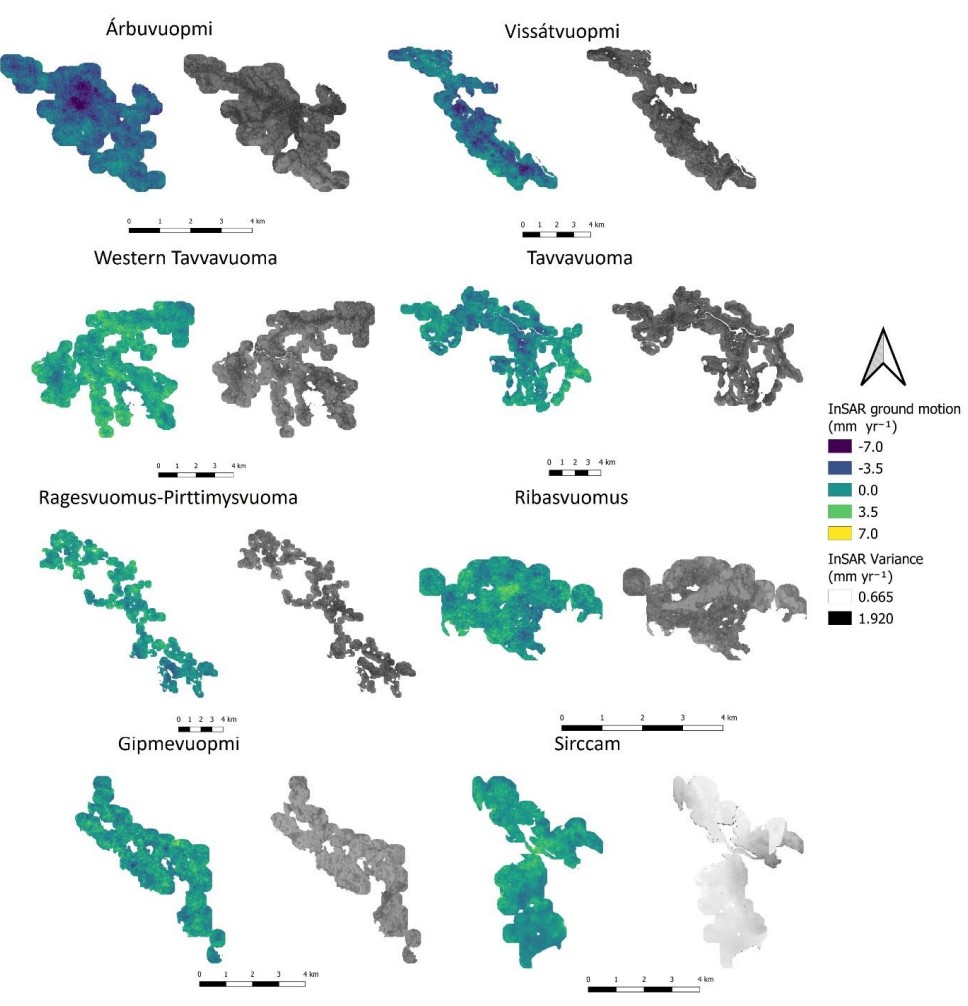

*Figure. 2: Palsa ground motion measured using Satellite InSAR showing differing levels of degradation across*
*the eight study sites. Sites are ordered by their latitudinal position. Negative values correspond to subsidence.*
*Note that in order to plot continuous areas the scenes shown are the palsa peatland area plus a 250m buffer*
*around each 100m ×100m pixel that cover a minimum of 1% palsa (Backe 2014). This means that areas of non-*
*palsa peatland and some areas with mineral soil are included in the figure. ASPIS InSAR variance were less*
*than 1.5mm yr$^{-1}$ in over 90% of pixels.*

Subsidence was recorded in just under half of the pixels all the eight palsa complexes (Table 2). Across the target
sites 3046.6ha (Table 2) out of the total site area of 5523ha (Table 1) were subsiding, which equates to ca 55% of
the total palsa complexes' area. Out of the subsiding parts of the palsa complexes, 405ha were subsiding at rates
>3.5mm yr$^{-1}$ at near gaussian distribution. However, it is evident from the frequency distribution plots, that it is in
the palsa complexes in the far north of the region that subsidence dominated the surface motion measured (Table
2, Figure 3). At Vissátvuompi and Árbuvuopmi 98 and 92% of the palsa complexes were subsiding with maximum
subsidence rates of -9.9 and -8.9mm yr$^{-1}$, respectively. The measured area affected by high subsidence rates of
between (>3.5mm yr$^{-1}$) were 204.8ha and 138.4ha at Vissátvuompi and Árbuvuopmi, respectively. This means
that ca. 30% of the total combined area of these two sites (1194ha) is in the highest range of subsidence. The high





degree of palsa subsidence at Vissátvuompi and Árbuvuopmi was confirmed by field observations at these sites
(Sofie Sjogersten, pers. Obs.): Both sites showed signs of active lateral erosions, large scale subsidence and
thermokarst formation. The more southerly sites also show subsidence, although rates were much lower, with the
-1 and 1mm yr$^{-1}$ range being most common (Fig. 3). Areas further to the south and west showed signs of uplift,
particularly the western parts of Tavvavuoma and Ribasvuomus with maximum rates of uplift of 6.3mm across
some smaller parts of these sites. However, all sites have some degree of subsidence, albeit at a lower rate
compared to the heavily subsiding northern sites.

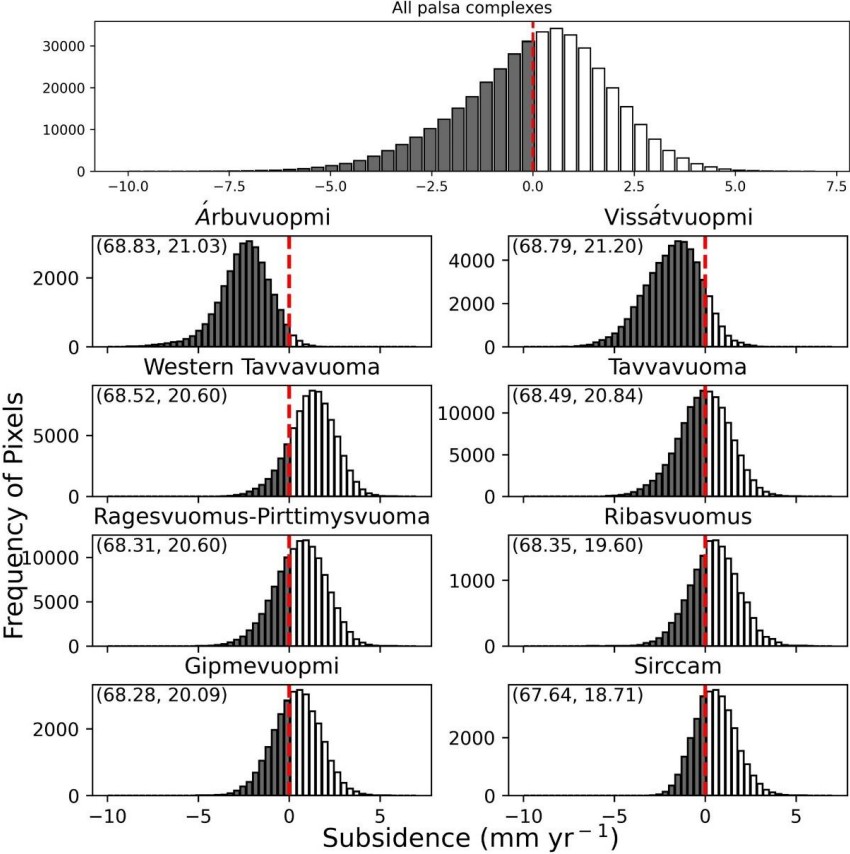

*Figure. 3: Distribution of 20m ×20m ASPIS InSAR pixels within each of the palsa complexes in this study and*
*the overall trend of the dataset according to the distribution of pixel moving in a particular direct and a given*
*rate. Shaded areas with negative values correspond to subsidence. The dashed central lines indicate pixels in*
*stable areas with no motion. Central point latitude and longitude is provided for each site in brackets for each*
*site.*

Calculating the roughness index from the DEMs at each palsa complex enabled differentiation of palsa from
surrounding lower lying and flat fen areas. Representative example complexes are shown in Figures 4 and 5 -
Vissátvuompi and Western Tavvavuoma, Overall, the palsa complexes to the north (e.g., Fig. 4b, c) display a
more pronounced topography across the focus areas than the more south-westerly ones (e.g., Fig. 5b, c). There
was clear correspondence between density of palsa and subsidence, i.e., areas with more palsa showed more
subsidence (Fig. 4a, d). Furthermore, the palsa complexes showed greater elevation variation compared to





surrounding fen areas and were more densely clustered to the north than in the more south westerly sites. These
features spatially coincided with higher subsidence. Substantial within site variability in subsidence was evident,
where the pixels with the highest subsidence rates being clustered together and following landscapes features,
e.g., palsa plateau edges. It was evident that many separate palsa complexes in an area resulted in a high degree
of elevation change, causing a high roughness index. In turn, areas with high roughness have the greatest
subsidence (Fig. 4,5). Visual comparison between orthophotos and roughness showed that areas of high roughness
corresponded well with areas of severe permafrost degradation (as indicated by lateral erosion and thermokarst
formation).

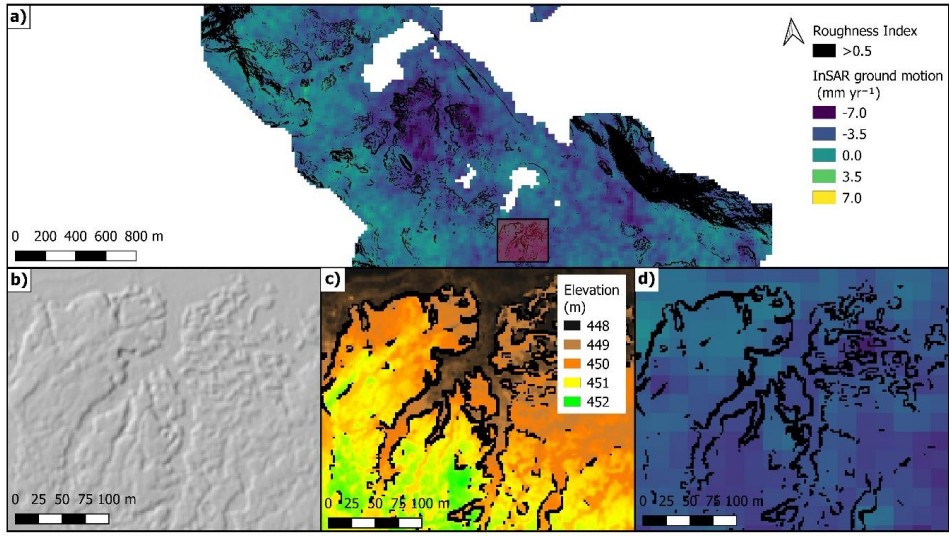


*Figure. 4: Visual analysis of Vissátvuopmi one of the sites where the most subsidence was found to be*
*occurring. Evaluation of correspondence of hillshade DEM (b), DEM (c) and InSAR subsidence (d) with Palsa*
*complexes suggested by roughness overlays. The positioning of b,c, and d within the larger site (a) show bands*
*of subsidence in proximal to roughness patches suggesting Palsa.*

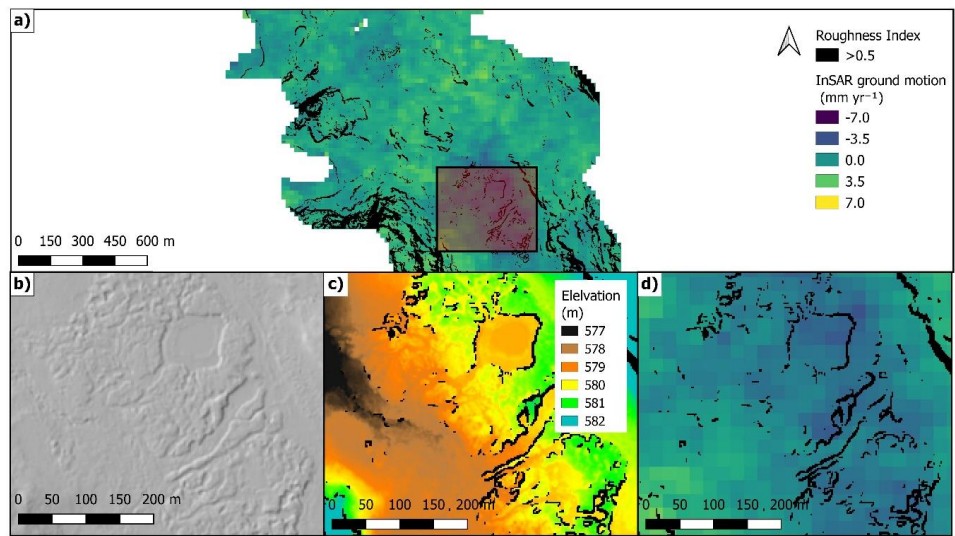


*Figure. 5: Visual analysis of Tavvavuoma which was found to have much lower levels of subsidence in*
*comparison to more northern sites. Evaluation of correspondence of hillshade DEM (b), DEM (c) and InSAR*
*subsidence (d) with Palsa complexes suggested by roughness overlays. The positioning of b,c, and d within the*
*larger site (a) show many less bands of subsidence and potential palsa than Figure 4.*

Regression analysis showed a relationship between roughness and subsidence as sites with greater subsidence
were also found to have greater roughness (Fig. 6a). Higher percentage palsa in a location was linearly related to
subsidence with the greatest subsidence found in areas with the highest percentage palsa cover (Fig. 6b). It was
also clear that the modelled permafrost probability did not correspond to the percentage of palsa, i.e. pixels with
100% palsa are in some instances predicted to have no permafrost (Fig. 6b).





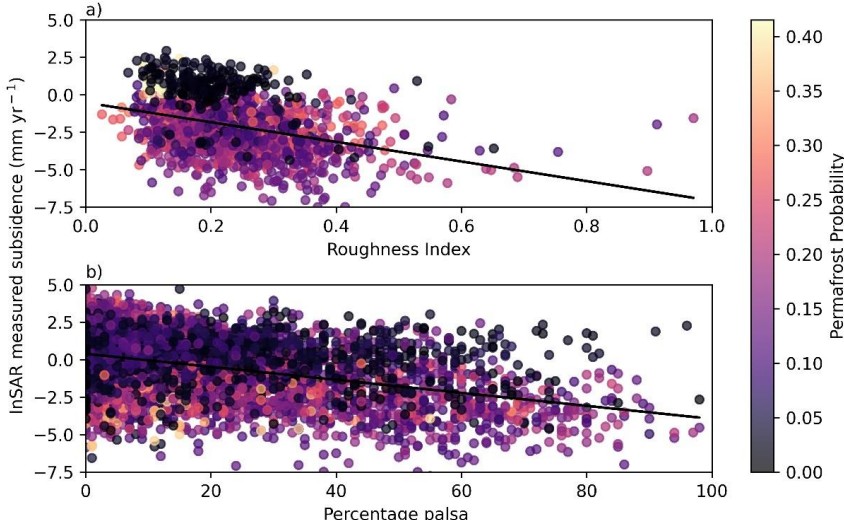


*Figure. 6: Relationship between a) the roughness index; p < 0.001, $R^2$ = 0.35 and b) percentage palsa in a*
*pixel; p < 0.001, $R^2$ = 0.41 and subsidence. The colours indicated for each data point are the analysed*
*probability (on a scale from 0 to 1) that an area would include permafrost, (Obu et al., 2018). Note that there is*
*less data for the analysis of roughness as the roughness was characterized only for the eight study sites and not*
*all palsa areas. Roughness values from valley sides (which at time were included in the buffer areas) are not*
*used in the figure.*

333

The analysis of the metrological data showed variability in both weather and climate across the study region in
part reflecting the patterns in the subsidence data. The warmest minimum and maximum temperatures, -29.2 and
32.8°C respectively, were recorded for the palsa complexes north of Lake Tornetrask, i.e. Gipmevuomi and
Ribasvuomus (Abisko weather station) (Fig. 1). The temperature in the area of Árbuvuopmi, Vissátvuopmi, and
Tavvavuoma palsa complexes (Saarikoski/Naimaka and Karesuando weather stations) ranged between -39.4 and
30.5°C (Table 3, Fig. 7a). The Katterjåkk weather station located in the mountains close to the Norwegian border
recorded the greatest annual snow depth measure of 229cm and a mean of 50cm. Note that in this far western part
of the study area palsa peatland were not present anymore. In contrast, the three other sites had comparable annual
snow depth with a mean of 20-30cm (Table 3, Fig. 7b).

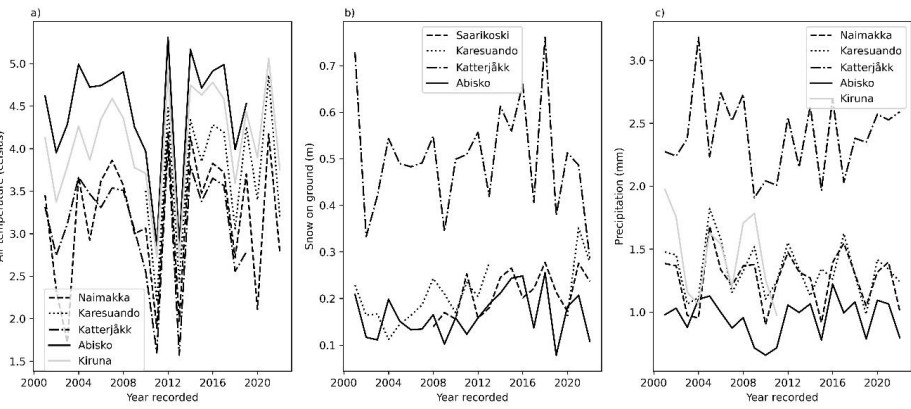




*Figure. 7: a) Mean annual daily maximum temperature, b) snow depth on the ground, and c) daily precipitation*
*at the meteorological stations in the study region (SMHI 2022).*
*Table. 3: Temperature and snowfall descriptive statistics. The snow depth data are estimated from days with*
*snow on the ground. Mean annual temperature and precipitation are averaged from 2000 to 2021. Maximum,*
*minimum and the inter-quartile range of daily maximum temperature and daily precipitation since 2000 are*
*also shown. Some weather stations lack certain years but were considered to have adequate coverage for this*
*task while two sites did not have sufficient data collection during the time period to be reliable and were shaded*
*out.*

| Weather Station | Temperature (ºC) | | | | Snow depth (m) | | | Precipitation (mm) | | |
|---|---|---|---|---|---|---|---|---|---|---|
| | Mean annual | Max daily | Min daily | IQR daily | Mean annual | Max daily | IQR daily | Mean annual | Max daily | IQR daily |
| Naimakka | -1.40 | 29.5 | -38.2 | 15.7 | | | | 456 | 50.8 | 1.0 |
| Saarikoski | | | | | 76.9 | 0.85 | 0.43 | 422 | 43.6 | 0.9 |
| Karesuando | -0.70 | 30.5 | -39.4 | 16.9 | 75.1 | 1.00 | 0.40 | 490 | 53.2 | 1.1 |
| Katterjåkk | -0.32 | 29.5 | -27.6 | 13 | 183.0 | 2.29 | 0.97 | 875 | 104.3 | 3 |
| Abisko | 0.53 | 32.8 | -29.2 | 13.5 | 60.0 | 1.27 | 0.42 | 348 | 61.9 | 0.6 |
| Kiruna | 0.06 | 30.3 | -30.6 | 15.6 | 5.3 | 1.13 | 0.45 | 545 | 53.1 | 0.9 |


There was no detectable difference in climatic trends among the meteorological weather stations since 2001 (p >
0.05). In contrast, the longer-term climate records show a strong increase in winter precipitation over the last 140
years at Karesuando, the northern most weather station ($F_{1,136}$=122.33, p < 0.001; $\chi^2$=47.0 %; Fig. 8a). This long-
term trend was also evident, albeit less strong, in Kiruna ($F_{1,110}$ = 28.17, p < 0.001; $\chi^2$=19.7 %; Fig. 8b). In Abisko,
the pattern of increasing in winter (DJF) precipitation was less clear ($F_{1,108}$=8.29, p < 0.01; $\chi^2$=6.3 %; Fig. 8c).
Snow depth, temperature, and summer precipitation (JJA) did not show clear temporal trends (data not shown).

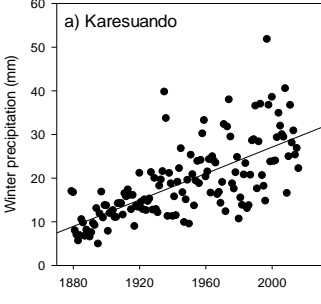 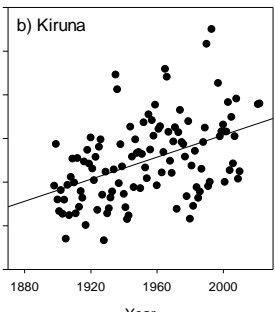 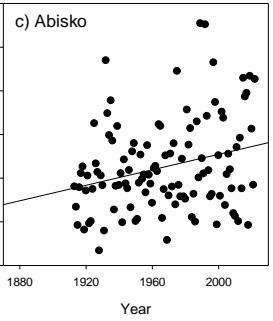

*Figure 8. Mean winter (DJF) precipitation over time at a) Karesuando, b) Kiruna, and c) Abisko, significant*
*trendlines are shown.*

## 4. Discussion

By way of satellite ASPIS InSAR-derived surface motion and associated spatial and statistical analyses, we
have demonstrated on-going, subsidence in the palsa peatlands of northern Sweden driven by a warming
climate. Based on the compelling agreement of subsidence with palsa landforms and their roughness, we
interpret this as permafrost degradation, i.e., thaw of the permafrost core within palsas and disintegration of



these landforms. This is in line with a wide range of literature (see introduction) and concurs with the local-scale
studies in the area undertaken using both satellite- and field-based methods (de la Barreda-Bautista et al., 2022;
Olvmo et al., 2020; Sannel, 2020; Sannel et al., 2016; Sannel & Kuhry, 2011), as well as with, the severe
climate warming impacts on temperatures and precipitation noted in the region (Hänsel, 2020; Irannezhad et al.,
2017; Vikhamar-Schuler et al., 2016) and the modelled predictions of total loss of permafrost across the region
within decades (Fewster et al., 2022). We suggest that the surface subsidence of the sample palsa complexes
measured in this study, together with complementary work in Norway (Borge et al., 2017), can be taken as
evidence of significant permafrost degradation in all palsa peatland areas across northern Fennoscandia.
The processes driving the degradation of the permafrost, as measured by the ASPIS InSAR-derived subsidence
data, are complex. Although permafrost degradation was observed in all the palsa complexes, rates varied both
within and among palsa complexes (Table 2, Fig. 2 and 3). Overall, the InSAR subsidence data demonstrates a
north to south gradient in increasing degradation. This indicates that local factors, such as local climate warming
responses or permafrost temperature, determine the sensitivity of particular areas and that regional climatic
gradients play a role in the long-term trajectory of these ecosystems (Johansson et al., 2011; Olvmo et al., 2020).
In particular, winter precipitation is generally considered a strong predictor of permafrost degradation due to the
highly insulating properties of snow, preventing heat dissipation during winter (Olvmo et al., 2020; Seppälä,
2011). This points to increased winter precipitation in the part of the northern most part of study areas as a
driver of the higher subsidence rates at the northern most palsa complexes (Table 2 and Fig. 7a). Interestingly,
climate data from the last two decades did not reveal strong differences in climatic conditions over the area. This
suggests that long-term trends combined with a buffered system reaction to change are driving regional patterns
in permafrost degradation.
It could also be the case that the observed north to south gradient of subsidence rates reflect different phases of
progression in an ongoing trend of permafrost degradation across the study region of northern Sweden. It is
plausible that the degradation process has progressed further at the more southern sites, reflecting higher
permafrost temperatures, and that as a result, subsidence rates have now slowed. All the while at the northern
sites, which still have a high cover of palsa: 26.3 and 31.6 % at Árbuvuopmi and Vissátvuopmi respectively,
show high subsidence rates. This is supported by research showing rapid permafrost degradation in the
southernmost palsa complexes in Sweden (Zuidhoff, 2002; Zuidhoff & Kolstrup, 2000) and in the area around
and to the south of Tornetrask, since the 1960's (Åkerman & Johansson, 2008; de la Barreda-Bautista et al.,
2022; Varner et al., 2022). However, permafrost degradation in palsa peatlands has progressed over longer-time
periods even in the far north of Scandinavia. Here palsas' have decreased in areal extent by 33– 71% over ca. 60
years, with more rapid contraction in recent years in Finmarkvidda, Norway and 54% in Vissátvuopmi, northern
most Sweden (Borge et al., 2017; Olvmo et al., 2020) and total loss of palsa complexes has been recorded in the
far north eastern parts of Norway (Vorren 2017).
Although there are differences in subsidence rates among sites the region wide permafrost degradation reflects
ongoing climatic trends (Fig. 2 and 6). Since 1901 Scandinavia's climate has become wetter as well as warmer
with a greater proportion of the precipitation falling as rain relatively to snow (Hänsel, 2020; Irannezhad et al.,
2018; Irannezhad et al., 2017; Vikhamar-Schuler et al., 2016). These trends are reflected in the far north where
higher air temperatures, greater precipitation and snow depths has already shifted climatic conditions, in parts of
the region, away from those that support permafrost in peatlands e.g. since the 1940's (Åkerman & Johansson,
2008; Borge et al., 2017; Olvmo et al., 2020).  Further, deep permafrost boreholes show decadal signals of
increasing temperatures in the Scandes mountains suggesting that warmer temperatures have been impacting
permafrost since the 1920's (Isaksen et al., 2007). Hence, is seems that climate warming has been impacting
permafrost in Scandinavia for at least 100 years.
As a result of the ongoing trend of increasing permafrost temperatures in palsa peatlands in Scandinavia, their
permafrost temperatures are now close to 0°C, making them very vulnerable to decay in response to further
increases in temperatures (Christiansen et al., 2010; Farbrot et al., 2013). Palsa formation is closely linked to the
mean annual temperature, with temperatures between -1 to -2°C over consecutive years needed as a threshold for
palsas to form (Vorren, 2017). In this context it is important to note that the MAT in the area was between 0.53
and -1.4°C since 2000 suggesting that at least in parts of the study area the climatic conditions do not support
formation of palsa anymore while conditions are marginal for palsa preservation in the entire region.



Although subsidence dominated in the northern sites, uplift was also noted in the study region. Mechanisms that may explain patterns of uplift are formation of new palsa as well as short-lived frost mounds that can form temporarily in the palsa system (Zuidhoff, 2002). Further mechanisms that may result in uplift are changes in the water level of the flooded parts of the peatlands as well as accumulation of plant residues from the productive fen vegetation parts of the study sites on the peatland surface, reflecting adaptation of the local ecosystemto degraded palsa mounds reflected by changes in remotely sensed terrain surface.

In addition to demonstrating regional permafrost degradation in northern Fennoscandia this work also provides proof of concept for circumpolar assessments of permafrost degradation using ASPIS InSAR. It enables detection of the areas with rapidly degrading permafrost and deepening active layers but also peat consolidation in areas that has already lost its permafrost (de la Barreda-Bautista et al., 2022). The fact that InSAR data is integrated over $20m \times 20m$ pixels means that the signal of local level degradation may be somewhat dampened (de la Barreda-Bautista et al., 2022). However, the high precision of the change in vertical position means that InSAR is an important tool to employ to detect the initial stages of large-scale permafrost degradation. Currently, the study of long-term trends and drivers using InSAR is somewhat limited by the short collection period of Sentinel 1, but as more data are continued to be collected, methods such as non-linear time series creation will become viable to compare subsidence directly to longer climatic drivers. However, the large scale baseline assessment of permafrost subsidence, developed here, provides an initial assessment of ongoing subsidence. would be advantageous should field monitoring be arranged in the future. As a complement to the ASPIS-InSAR data, the novel roughness thresholding method used here together with contextual data proved a powerful tool to map and monitor changes (Franklin, 2020; Konig et al., 2019; Otto et al., 2012). This approach could be developed using machine learning methods to model palsa dynamics to better automate the extraction of palsa landform positions (Konig et al., 2019; Luoto & Seppälä, 2002). If accomplished, the operating extent of this tool could be vastly increased using the Arctic 2m DEM dataset over area were its quality is high enough to allow high resolution mapping of the degrading edges of raised palsa plateaus (Morin, 2016). Together the ASPIS-InSAR and the DEM derived roughness index metrics offer novel ways of large scale monitoring of permafrost degradation. This will help to quantify the rate of palsa ecosystem collapse and transition to a non-permafrost state.

We conclude that permafrost degradation of palsa peatlands is occurring across northern Sweden, with the greatest rates of degradation and largest areas impacted being Swedens two largest permafrost peatland complexes in the far north. This raises serious concerns that these systems will lose their permafrost entirely in the coming decades especially as climatic conditions are approaching the limits of sustaining palsa peatlands (Fewster et al., 2022). The implications of this rapid loss of permafrost is ecosystem collapse and loss, as the permafrost core is fundamental to the existence of palsa peatlands. Future research should focus on the implications of this collapse on increased $CH_4$ emissions (Glagolev *et al.*, 2011; Turetsky *et al.*, 2020; Varner *et al.*, 2022),carbon loss (Hugelius *et al.*, 2020), and thus the potential for strong climate feedbacks (IPCC, 2021) as well as using longer-time InSAR data as this becomes available to investigate regional variations in climatic drivers of permafrost degradation. Further, our study demonstrates that InSAR together with terrain data can be applied over continuous natural surfaces at a regional-scale to monitor permafrost degradation in palsa peatlands, offering a tool for circumpolar monitoring of climate warming impact on these systems.

## 5. Acknowledgement

This work was supported by funding from the University of Nottingham, UK, EU-InterAct funding via the InterAccess programme and the Swedish research council (VR-2021-05767 to M. Siewert). Associated fieldwork was supported by the Climate Impacts Research Centre (CIRC) at Umeå University. Samuel Valman was supported by the EPSRC funded Geospatial Centre for Doctoral Training (EP/S023577/1).

## 6. Author contributions

SV: Carried out the majority of the data analysis and made a significant contribution to data interpretation, writing and finalising the manuscript text. Both SV and MS can be considered to have contributed equally to this work.



MS: Contributed to the conception of the study, contributed DEM and orthophoto data, carried out fieldwork to
assess permafrost degradation, contributed and advised on data analysis and interpretation, contributed to struc-
turing, writing, and refining the text. Both MS and SV can be considered to have contributed equally to this
work.
DB: Contributed to the conception of the study, advised on the data analysis, and made a significant contribution
to finalising the text.
ML: Provided data analysis, support on the InSAR processing, data interpretation, and writing of the text.
DG: Carried out the initial InSAR data processing
BBB: Contributed to the conception of the study and refining the text.
AS: Contributed to the conception of the study and advised on the InSAR data processing
SS: Conceived and directed the study, contributed to data analysis, carried out fieldwork to assess permafrost
degradation and made a significant contribution to formulating and finalising the text.
SS, DB, AS and MS secured the funding for the project.
**Code Availability**
All the python scripts used to carry out these analyses are available at the github repository:
https://github.com/SamValman/Permafrost_Sweden.
**Data Availability statement**
The Sentinel-1 datasets are freely available and can be obtained by searching and downloading the Interferomet-
ric Wide (IW) swath mode products for orbit track numbers ?? and ?? through the Copernicus Open Access Hub
(https://scihub. copernicus.eu/dhus/#/home). The processed interferometric data and deformation maps are com-
mercially sensitive and may be made available on reasonable request by email addressed to the corresponding
author. All other datasets produced during this project will be uploaded on zenodo and the DOI provided once
the article has been accepted.

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
