# Peer review of "InSAR measured permafrost degradation of palsa peatlands"

_The Cryosphere, 2023_

## Author Response (AR1)

Nottingham Geospatial Institute

University of Nottingham

Nottingham

NG7 2TU

2024/02/07

Dear Dr. Lee and Reviewers,

Thank you for the time and effort that you have put in to reviewing our manuscript. The revisions have been valuable for framing our article in the best possible way. Both reviewers have asked for more information about the InSAR methodology which we have now provided in the text along with some additional references to the methods used. The majority of the minor comments revolved around phrasing and grammar. As requested by the reviewer, we have carefully re-read the article and made definitions of palsa and palsa complexes consistent throughout the manuscript.

Reviewer one provided some suggestions which were directly incorporated into the article, we thank them for putting the effort in to do this. The reviewer also recommended the inclusion of Orthophotos in the document, which we have now done. Reviewer Two also provided some grammatical points which we have acted upon and thank the reviewer for these. They identified the need to be more precise with when the orthophotos and DEM tiles used were captured, which we have now done. This should make it easier for readers to assess our results.

We believe that these final modifications have further strengthened our manuscript, and hope that it can now be considered ready for publication.

Yours sincerely,

Samuel Valman

**Itemised responses**

**Response to Review One:**

Thank you for the time and care you have taken reviewing our manuscript, the suggestions made have helped provide a significant improvement to the Manuscript. We have now carefully edited the manuscript to remove grammatical errors as suggested. Below we also provide an itemised response to the individual comments.

Comment 1: *Consider adding InSAR in the title, as it is the only dataset on which you base your investigation of permafrost degradation.*

Response: We agree this would help direct readers to the manuscript, whilst we had originally focused the title on the target rather than the sensor, this may provide a wider audience and may better fit with positive outlook for this method in the future. The title has been changed to "InSAR measured Permafrost degradation of peatlands in northern Sweden".

*Comment 2: L 40: landscapes -> land area or exposed land surface*

Response: We agree with the reviewer's suggestion and have changed this sentence to read "exposed land surface". This can now be found on line 40.

*Comment 3: L 41–42: "…of which around 1.7 million km2 is permafrost substantially overlapping with the circumpolar permafrost region" -> perhaps just: "of which around 1.7 million km2 is affected by permafrost" or ".. is within the permafrost region"*

Response: Thank you for the suggestion, we have simplified the text to "within the circumpolar permafrost region." On line 42.

*Comment 4: L 43: I suggest specifying that this applies mainly to peatlands in discontinuous and sporadic permafrost zones.*

Response: This has been added to the end of the sentence on line 43, so the reader is not given a false impression of the scope of permafrost regions.

*Comment 5: L 44–46: Please rewrite this part. For instance, palsa peatlands in Sweden as an example in a separate sentence.*

Response: We understand that the wording made this difficult to understand and have therefore rewritten the second half of the sentence into a new sentence on line 46, such that "In northern Sweden, 137 km2 of these palsa have been recorded from field reports (Backe, 2014)."

*Comment 6: L 50: Unsuitable permafrost conditions where? Everywhere? Fennoscandia? Please specify.*

Response: Although conditions may become unsuitable in many areas the most pessimistic estimate of 2040 was from Fewster et al. (2022), which was about Fennoscandia in particular. Therefore, we have made this clear in the text that in this case we are referring to Fennoscandia on line 52.

*Comment 7:" L 58: "…frozen permafrost core of palsas…" -> e.g. "perennially frozen core of palsas" or just "frozen core of palsas""*

Response: The sentence in question is regarding winter conditions that impact year round permafrost and as such I have changed this to perennial frozen core of palsas on line 59.

*Comment 8: L 68: Abrupt thaw is not limited to peat plateaux but can also occur at the edges of palsas. Also, earlier on L 44 you use 'plateaus', not 'plateaux'. Either form is fine but be consistent.*

Response: We have made the change from plateaus to plateaux throughout the document. And we also include palsa edges to the sentence about abrupt thaw on line 68.

*Comment 9: L 118: insert 'data' after 'LiDAR'*

Response: We have made this change on line 119.

*Comment 10: L 126–127: Your use of 'palsa' and 'palsa peatland complex' here is confusing. Could this be rewritten as e.g., "The eight largest palsa peatlands in the region range between 50 …" ? Also, please clarify what do you mean by 'palsa peatland', 'palsa peatland complex' and 'palsa complex' in your paper. If they are interchangeable, then use only one.*

Response: We agree with the reviewer that these terms are nearly interchangeable and create confusion for the reader. We have changed the terminology to just palsa(s) unless it is a palsa complex which has been used when we are exclusively referring to one of the eight larger palsa sites where there is a concentration of palsa. In the introduction we already have explained that palsa is a form of peatland so repetition of this would not be required. And the term complex is explained in the text "hereon in referred to as palsa complexes, a term reflecting their mosaic nature of raised palsa plateaux" on line 142.

*Comment 11: Figure 1: Please export this figure in better resolution, e.g. as PDF. What does 'ungrouped palsa' mean in the legend? My guess is that it refers to the scattered black dots representing palsas, which were not part of this study. Could they be indicated with some different colour/symbol to differentiate from the peatlands, which you do focus on?*

Response: Thank you for this comment, ungrouped palsa are the palsa raster cells from the Backe (2014) report that are not part of the eight palsa complexes we focused on in this study. While we focused on the eight large palsa complexes, we did not want to totally exclude this useful data and having used it in the rest of the study we included them here. Although this is later clarified in the text "surface motion at these eight palsa complexes, and over all individual palsa peatland raster cells in the region, were produced", we have changed the text in the figure caption on line 134 to read "The black regions show all the palsa which has been reported to exist (Backe, 2014), with the larger named areas displaying the 250m buffers around the palsa areas which have created continuous expanses."

*Comment 12: L 140–149: I think this could be rewritten to be more concise (e.g. avoid repetition of the fact that the eight selected areas are the largest concentrations of palsas in Sweden) and easier to follow. The last sentence of this paragraph is particularly troublesome. Also, please clarify, did you create the 250 m buffers or was this done already by Backe (2014).*

Response: We have rewritten this paragraph on line 138 to "A previous national palsa mapping dataset provided raster cells at a spatial resolution of 100m, with the % palsa cover computed and a 250m buffered output to provide continuous palsa area outputs (Backe, 2014). This afforded analyses at a spatial resolution suitable for analysis with Sentinel-1 yet provide practical representation of the condition of the palsa in this region. All these data were analysed in this study, but the eight largest continuous areas of these palsa (Backe, 2014) were focused on, hereon in referred to as palsa complexes, a term reflecting their mosaic nature of raised palsa and/or peatland plateaux, interspersed with lower lying fen or thermokarst areas. These eight sites account for the majority of the palsa areas in Sweden, the sites are listed in Table 1 along with some associated information on

their status and total and raised palsa plateaux areas." Which also helps clarify that we analysed all the data but focused mainly on these complexes, thus helping to answer the later comments that it was easy for a reader to misunderstand that we also carried out analysis on the wider palsa raster cells.

*Comment 13: L: 145: "raised palsa plateaux" -> e.g. "raised palsas and/or peat plateaux"*

Response: We have made the change as recommended on line 143.

*Comment 14: Table 1: 'EU Nature 2000' -> 'EU Natura 2000'. I also suggest adding a column showing the respective weather stations, from which the meteorological data were used.*

Response: Thank you for noticing the misspelling in 'EU natura 2000' we have amended the table to reflect this. We agree that the weather information is important but because we do not directly tie it to individual raster cell responses, we would argue that the positions of weather stations in figure 1 is easier for the reader to visualise and therefore sufficient in this study.

*Comment 15: Please clarify, which areas were included in what analyses. I understood that you only focused on the eight palsa peatland complexes in your InSAR ground motion detection and other analyses. However, in some cases (e.g. L 248–251) it sounds that your InSAR analysis covered larger areas than e.g. roughness index.*

Response: We mainly focused on these sites but where possible and useful we also utilised the full palsa raster dataset available to us. This is described in the text, but we believe by replying to comment 21 we have made this much clearer to the reader by grouping the analysis into sections which better define which datasets were used.

*Comment 16: Overall, I would like to see more detailed description of data processing. At least add a flowchart indicating different procedures, for which steps 100 m spatial resolution was used etc. Keep in mind that not all readers of the Cryosphere are experts in InSAR. I think with such flowchart you can also easily address my previous comment.*

Response: We thank the reviewer for this suggestion. As recommended, we have now included a flowchart outlining the data processing steps for the InSAR subsidence mapping on line 188. We hope this satisfies the reviewer's request for a more detailed and accessible description of data processing for readers who are less familiar with InSAR.

*Comment 17: What is the vertical accuracy of the InSAR data you use? How much of the detected ground motion is within the margins of error? Perhaps this is something that could also be addressed in the discussion.*

The vertical accuracy of the InSAR data is best represented using the standard error. The standard error for each palsa complex has now been included in a new column titled 'Mean standard error (mm yr$^{-1}$)' on line 270 and we have discussed the implication of this within the discussion on line 451.

We conclude that because 69% of ground motion rates are within the margins of the mean standard error, there is greater confidence in the direction of surface motion as opposed to the magnitude. We hope the reviewer is satisfied with this interpretation.

*Comment 18: L 179: What is SNAPHU algorithm and how did you modify it?*

The SNAPHU algorithm (Chen and Zebker, 2002) is a solution to the phase unwrapping problem which is a major problem for any InSAR algorithm. Essentially, it is only possible to measure interferometric phase modulo 2π and an unwrapping algorithm attempts to solve this by adding appropriate multiples of 2 π to the phase to produce a linear measure. The SNAPHU algorithm was modified in order to speed up the process for larger areas. Modifications included a complete re-engineering of the code in order to allow the ability to parallelise and to spread the calculation across multiple cores. We have now included the (Chen and Zebker, 2002) reference in the text on line 179 and state what the modifications are for.

Chen, C.W. and Zebker, H.A., 2001. Two-dimensional phase unwrapping with use of statistical models for cost functions in nonlinear optimization. JOSA A, 18(2), pp.338-351.

*Comment 19: L 223: "The Naimakka station did not provide" and "the Saarikoski station did not provide" sounds as if the data were there, but the stations did not want to share it, which I doubt. Please rephrase.*

Response: Thank you for this suggestion we agree the original text infers a different meaning to the that which we intended. To rectify this, we have changed the text to the stations "did not record" on line 240, to make it clearer.

*Comment 20: L 244–245: Which time series were incomplete and how short time series were used in those cases?*

Response: We agree that this is information that the reader needs to know, however we feel the best way to convey it is to direct the reader to Figure 8. Where they can see which time series have gaps and where. As such, we have added "(See fig. 8)" on line 268 to this section.

*Comment 21: I suggest dividing the results section into two or three subsections to make it easier for a reader to navigate through the paper. Same applies to the description of datasets and analyses.*

Response: Thank you for the suggestion, along with some changes suggested by reviewer two we have re-organised the analysis section of the methodology into 2.3.1 Surface motion statistics, 2.3.2 Roughness thresholds, 2.3.3. Causes of surface motion, and 2.3.4 climatic factors. In the results section we have followed a similar sub-section structure as suggested.

*Comment 22: L 248–251: Does this mean that you derived surface motion from other palsas than the eight focus areas as well? Or where did this 7.7 mm yr-1 uplift occur?*

Response: The reviewer is correct, we derived surface motion for the entire area of Sweden, we then clipped this to all the palsa raster cells provided by the Backe, 2014 study. We have explained this in the methodology text: "From this the frequency distributions of ASPIS InSAR surface motion at these eight palsa complexes, and over all individual palsa peatland raster cells in the region, were produced." On line 209, but agree that it is not clear enough in the results section and therefore have added the additional text: "raster cells measured in northern Sweden" on line 266, which will make a clearer distinction with the next sentence that reads "Focusing solely on the eight palsa complexes… (Table. 2.)"

*Comment 23: Figure 2: Could you provide larger versions of these plots alongside respective aerial orthophotos and (estimated) edges of palsas? If not in the main text, at least as supplementary material. In the current form it is impossible for the reader to evaluate, which parts of the plotted areas are actual palsas, and how the InSAR ground motion values are distributed in relation to the permafrost areas.*

Response: We have made these plots page size to make it easier to see where there is ground motion in these sites. We provide a version of the plot with the 100*100m raster cells overlaid; however, we feel that this makes the image harder to read and so have only included it in the supplementary material. We have also included some examples of the orthophotos and the ground motion at the same sites in the supplementary.

*Comment 24: L 271: "just under half" is very vague. Also, this seems to contradict with the next sentence, where you say that ca. 55 % of the total palsa complexes' area was subsiding. Please clarify.*

Response: After consideration, we agree that this sentence is vague and is not required to move the results section forwards and therefore has been removed. We will only keep the 55% statistic which we have calculated.

*Comment 25: L 277–278: Do you mean "subsidence rates > 3.5 mm yr-1" or between 3.5 mm and some other value?*

Response: In this case, this is just greater subsidence than >3.5 mm per year, to display the areas that are subsiding the most. There are no areas which have lost more than 10mm therefore we did not see the need to develop an upper limit on this. The confidence we have in the potential for error, from answering comment 17, further supports this.

*Comment 26: Figure 3: Please change the title of the x-axis to "ground motion", as you have correctly called it in Figure 2. As you describe in the caption, only the shaded areas represent the subsidence.*

Response: That is correct thank you for pointing this out, we have made the suggested changes on line 302.

*Comment 27: L 306–308 and Figures 4 & 5: Here again, it would be good, if you could add aerial orthophotos as well as estimated palsa edges so that it is easy for the reader to compare different datasets. Regarding the figures, I advise changing the colour assigned to the roughness index or to the lowest elevation values. Same (black) colour is confusing particularly in Figures 4c & 5c. I also suggest removing the shading of the rectangles indicating the spatial extents of the close-ups in Figures 4a & 5a. Some coordinates would also be good. It would have been interesting to see, where in the landscape of Tavvavuoma those positive ground motion values are located, but unfortunately I could not locate, which part of the Tavvavuoma is illustrated in Figure 5.*

Response: Thank you for your thoughts on these figures. We have removed the shading as suggested but changed the box outline color to make it easier to see. We have also made amendments to the colormaps so that the blacks do not overlap. We have added co-ordinates to the 5a and 6a part of the figures so that it is possible to place these in the wider context of northern Sweden.

We agree the inclusion of aerial imagery would be useful here and we have included these as 5 and 6 e.

*Comment 28: Additional comment regarding Figure 4a: It seems that InSAR ground motion data depicts very nicely the subsidence of higher palsas ca. 800-1000 m north-west from the area, which you chose to show in the close up.*

Response: Thank you for the observation we have drawn the reader's attention to it in the figure caption on line 323 as a larger scale example of what we are trying to show with the close ups.

*Comment 29: L 319: What do you mean by "many less bands of subsidence and potential palsa"?*

Response: Here, we were trying to explain the visual arrangement of what we suspect is palsa in the image by describing it as a band of palsa. To make this clearer we have explained bands to ""bands" (linear arrangements of palsa across the image)" on line 332.

*Comment 30: Figure 6. Here again, please change "subsidence" in y-axis title to "ground motion".*

Response: Thank you for pointing out these errors we have corrected the y-axis title and checked all images for the same problems.

*Comment 31: L 331: What do you mean by "all palsa area" here?*

Response: We have clarified this sentence to "all palsa raster cells from (Backe, 2014)." On line 345. To make it clear that we are referring to all the other palsa that the study has measured with InSAR ground motion but is not included in the 8 complexes.

*Comment 32: What is your reasoning to include data from Katterjåkk station, if there are no palsa peatlands and it is not the closest station to any of the eight study sites? Since you do not refer to it*

*anywhere in the discussion, I suggest leaving it out. For example, in the figures 7b & 7c, the values from Katterjåkk draw the attention, while perhaps the records form other stations are more relevant for the study sites that you focus on.*

Response: Thank you for the suggestion we agree that because the weather data from this station does not significantly drive the discussion forwards it has been removed from the paper. We have removed it from all the figures including figure 1.

*Comment 33: L 357: This analysis was not mentioned in the data analysis section. Please describe there, which test you used, as well as the level of significance, which you use in all of the statistical analyses.*

Response: For all analysis, an alpha value of 0.05 was used to test for significance, this has been added to the methodology on line 234. This used linear regression analysis, with the year as the independent variable and the respective weather data as the dependent variable. The assumptions of normality and homogeneity of variance of the residual were assessed using residual plots in Genstat (VNS Ldt). This has been expanded on and clarified in the text on line 259.

*Comment 34: L 362: I think it would still be useful to see, how the air temperature, snow depth, and summer precipitation have varied over this long time period. If not in the main text, perhaps in the supplementary materials.*

Response: We agree that this would be useful for the reader who has further interest in the study, and we have added this figure to the supplementary materials.

*Comment 35: L 373–378: Unnecessarily long sentence.*

Response: This has now been separated into two sentences starting on line 384. One pertaining to smaller scale studies in the area and the other to the expected and modelled changes based on climatic conditions. "This is in line with a wide range of literature (see introduction) and concurs with the local-scale studies in the area undertaken using both satellite- and field-based methods (de la Barreda-Bautista et al., 2022; Olvmo et al., 2020; Sannel, 2020; Sannel et al., 2016; Sannel & Kuhry, 2011). The findings also agree with what is expected from the severe climate warming impacts on temperatures and precipitation noted in the region (Hänsel, 2020; Irannezhad et al., 2017; Vikhamar-Schuler et al., 2016) and the modelled predictions of total loss of permafrost across the region within decades (Fewster et al., 2022)."

*Comment 36: L 379–380: I do agree that permafrost degradation likely affects most of the palsa peatlands of However, to say that the degradation is "significant" and "…in all palsa peatland areas across northern Fennoscandia" based only on your study, in which some of the sites showed mostly uplift rather than subsidence, and one study from Norway, thus completely ignoring Finland and Kola Peninsula, overly simplifies the variety of dynamics in palsa peatlands in this region. I strongly advise to rewrite this statement.*

Response: We accept that this may have been overreaching our results and thus have changed the sentence to "can be taken as evidence of substantial permafrost degradation in many palsa areas

across northern Sweden and therefore, likely to be also occurring across northern Fennoscandia." On line 390. To reflect that the subsidence we have found in 55% of the study is fairly substantial especially on the northern sites and we would be worried for the state of palsa in northern Fennoscandia, which we accept may have different local variables but will also be suffering from climate change and so is likely to have similar issues.

*Comment 37: L 381: subsidence -> ground motion. Same on L 383.*

Response: We have changed the second part of these sentences on line 295 to read "although ground motion rates were much more stable, with the -1 and 1mm yr-1 range being most common". This specifies the difference between subsidence which we started the sentence with because we were talking solely about negative ground motion and so would prefer to continue using the term subsidence.

*Comment 38: L 384: Wasn't the gradient of increasing degradation/subsidence from south to north? Please check and correct if needed.*

Response: Thank you for pointing this out, we had the sentence reversed and have now fixed it to show that subsidence is more common to the north and less common to the South. Now found on line 396.

*Comment 39: You connect the InSAR-based ground motion results solely to the climatic variables. However, you do not provide any close-ups to the climatic conditions at the closest weather stations during the period covered by InSAR data (2017–2021). Were summer and winter air temperatures, summer and winter precipitation, and snow depths all similar at the northern and southern weather stations during this period?*

Response: We have calculated this and provide a figure in the supplementary material of these seasonal patterns over the 5 year period. However, Figure 8 provides temperature, snow cover, and precipitation from 2000 to 2021, which covers the same period as the InSAR data. It is possible to reduce this to just the 5 years of the InSAR data however we feel that shows no new additional data which is not visible from the current figure 8 but will reduce the ability to see the proximal years of climate which are likely to heavily impact the palsa formation and degradation that we are measuring here.

*Comment 40: I think you are missing the possible effect of palsas' shape and geometry on their degradation rates. This has been pointed out by, e.g. Borge et al. (2017) and Mamet et al. (2017), which you cite, as well as more recently by Wang et al. (2023; https://iopscience.iop.org/article/10.1088/1748-9326/ad0138). Your DEM and roughness index data in addition to aerial orthophotos seem sufficient to assess the differences in palsas' geometries between the sites. See also comment #28. I do not expect you to quantify all palsa geometries for this case, but at least some visual assessment would be beneficial for the discussion.*

Response: Thank you for this suggestion which had been discussed but not implemented, we think that the roughness index used here would be a good way for future studies to focus in on this

concept. Here we have discussed it and provided our view using figures 5, 6 and the orthophotos we have included in the supplementary material. The following discussion paragraph has been included on line 466: "It has been suggested that small, fragmented, and irregularly shaped palsa are more susceptible to erosion (Borge et al., 2017; Mamet et al., 2017, Beer et al. in review). We have not gone as far as to estimate this here but the possible palsa edges inferred from the roughness index, could be built upon for this understanding. Casual analysis of figures 5, 6 and the orthophotos provided in the supplementary materials would support these expectations." The Beer et al. paper was found through the Wang et al. 2023 reference provided by the reviewer, so we also thank the reviewer for bringing this to our attention.

*Comment 41: L 403: Here you use 'Scandinavia', as well as later in the discussion. Please clarify, whether you aim to discuss palsa peatland changes only in Scandinavia or in the whole northern Fennoscandia. These terms are not interchangeable.*

Response: We agree this needed thinking about, and we have replaced the discussion comments with Fennoscandia on line 392 because we believe the findings we have made here, relate to our knowledge of palsa in the whole region.

*Comment 42: L 419–421: I suggest adding the importance of thin snow cover, as is also mentioned by Vorren (2017), which you cite here. I also want to note that the palsas investigated by Vorren are located in climatically very different area.*

Response: We have changed this sentence to include the effects of insulating snow cover, line 432. We agree with the point about very different climatic conditions.

*Comment 43: L 424–429: I think the high number of pixels with positive ground motion values in your study areas deserves a bit more attention. As I mentioned earlier, showing aerial orthophotos and estimated palsa edges with your InSAR ground motion values could help evaluation of what could be the reason(s) behind the "uplift". Are positive values mostly in the fen areas? Or areas with tall vegetation on mineral ground?*

Response: There are several mechanisms responsible for driving the uplift seen in these images, we have discussed this in the paragraph on line 437. Delineating the palsa edges from the orthophotos is beyond the scope of this paper, but we have planned follow up studies to look into this. Instead, here we propose using the palsa percentages from the Backe (2014) study to compare 100*100m squares that contain no palsa to squares that do, within the eight buffered palsa complexes. This will enable us to separate if positive ground motion is a result of changes in palsa height or surrounding ground heights.

*Comment 44: L 439–441: I suggest editing this to, e.g.: "However, our large-scale assessment of permafrost subsidence provides a baseline for future assessment of subsidence in northern Sweden". In the current form the sentence is very difficult to read. And what "would be advantageous"? Your assessment for the future field monitoring or the field monitoring itself? I believe both.*

Response: We thank the reviewer for the suggestion which we have taken on board and have changed the sentence on line 456 to "However, our large-scale assessment of permafrost subsidence provides a baseline to direct, and compare against, future fieldwork monitoring in northern Sweden."

*Comment 45: L 518: Add line change before Ballantyne.*

Response: We have added in this line change.

**Response to Review Two.**

Thank you for the insightful comments, we believe in addressing these the manuscript will be significantly improved. There are a few grammatical points which have been made and as a result we have thoroughly read and revised the manuscript to remove errors and make the manuscript easier to read. Below we have provided an itemised list of the suggestions made and our responses which we hope you will agree, have helped improve the manuscript and make it more accessible to readers.

**Comments on Substance.**

*Comment 1: In the Introduction it would be interesting to mention what the current annual degradation rates are, if that information is available. Probably in the paragraph that ends around Line 95.*

Response: We thank the reviewer for mentioning this, in fact the difficulty in finding these data was one of the reasons that we set about to use InSAR data to provide a potential answer here. We have made the difficulties in finding degradation statics clearer in this paragraph (line 79) and include the references (de la Bautista Barreda 2022 and van Huissteden et al., 2021) which provide some indication of what these values might be in the area. These are then built upon with our findings here.

*Comment 2: The Introduction could use more info on what InSAR for subsidence studies have found and the accuracy they obtained.*

Response: Widespread conclusions on the accuracy of previous InSAR studies is that there is greater confidence in the direction of motion rather than the absolute magnitude. We have now included this comment within the introduction to summarise the accuracy of previous InSAR studies on line 100. The references for this have not changed (Alshammari et al., 2020; Alshammari et al., 2018; Bartsch et al., 2016; de la Barreda-Bautista et al., 2022; Short et al., 2014; van Huissteden et al., 2021).

*Comment 3: Line 130 – why not use Naimakka instead of Karesuando weather statistics? Most of the palsas that earlier existed in Karesuando have thawed and disappeared, so it doesn't represent the climate as well as Naimakka.*

Response: We agree with the reviewer that the climate at Naimakka would have been a better comparison however the time series available is much shorter. It starts in 1944 for temperature and precipitation and in 1961 for snow depth. We did find some palsa in the Karesuando area during the field validation carried out in this study, therefore we argue that although Karesuando is not representative of the sites as a whole it provides information on the edges of our palsa ranges.

*Comment 4: Line 142 – I would call it a grid cell or raster cell instead of a pixel, since a pixel refers to a picture element, usually from an image, while with a processed raster data set, it should be a grid cell or raster cell.*

Response: We have replaced these instances with the word raster cell to match the data type used in this section of the study.

*Comment 5: Under Section 2.2 Datasets, please put the processed cell size of the data, which should be 5 m in range and 20 m by azimuth if I'm not mistaken. Is this the resolution of the input data you worked with?*

Response: Thank you for pointing this out. We have now corrected the dimensions in Section 2.2 to 5 m in range and 20 m by azimuth on line 171.

*Comment 6: Line 166 – What program or code was used to run APSIS? Please describe more about APSIS, for example, I believe it needs to resample the result to be larger than the input pixel size, so I wonder how it is that your output is 20 x 20 m. This can help us interpret the result better.*

Response: APSIS was implemented using code written in-house by Terra Motion Limited.  It is based upon an algorithm formerly called 'ISBAS' which was first detailed for Sentinel-1 data in Sowter et al. (2016).  The current version of APSIS averages pixels from the Sentinel-1 IW product using a multilook window of 7x2 pixels to achieve an approximate final resolution of 20m across the site. See lines 161 line and 171.

Sowter, A., Amat, M.B.C., Cigna, F., Marsh, S., Athab, A. and Alshammari, L., 2016. Mexico City land subsidence in 2014–2015 with Sentinel-1 IW TOPS: Results using the Intermittent SBAS (ISBAS) technique. International journal of applied earth observation and geoinformation, 52, pp.230-242.

*Comment 7: Line 184 – what kind of accuracy? This sentence is a bit vague. Of identifying that permafrost degradation has occurred? Or accuracy of measured vertical subsidence?*

Response: We have changed the sentence to "have verified the ability to use InSAR as a tool to monitor permafrost degradation (de la Barreda-Bautista et al., 2022)." On line 185. To make it clearer that InSAR is viable for project carried out here.

*Comment 8: Line 189 – Are you sure you used panchromatic orthophotos? The ones available from 2016 are either RGB or False-color IR, so I would think you used one or the other of these, not panchromatic.*

Example response: Thank you for pointing this out we have checked and the RGB data were used. We have referenced the Lantmateriet orthophoto document to explain which data were used. The image data is largely from 2021, with two sites using imagery taken in 2018. This has been added to the text at line 196.

*Comment 9: Line 192 – The Swedish national lidar scanning consists of several different dates of data from ca 2009-2015, so it is likely not from 2016. You need to download the shapefile to determine the dates of the lidar data, which were acquired in 25 x 50 km blocks. See for example Nilsson et al, (https://doi.org/10.1016/j.rse.2016.10.022) for a description of the national lidar data, or else go directly to Lantmäteriets website for a description, as well as the shapefile.*

Response: The dataset we used was GSD-Höjddata, grid 2+ (which is made up from data collected between 2009 and 2019: (https://www.lantmateriet.se/sv/geodata/vara-produkter/produktlista/laserdata-nedladdning-nh/)). We have gone through the tiles that we used and replaced the date with a list of exact dates for different LiDAR sections in table 1, these are between 2013 and 2018.

*Comment 10: Line 201 – For the Discussion: How do you think taking the mean value over a 100 m x 100 m area may have affected your results? I am wondering why you didn't maintain the 20 x 20 m resolution from ASPIS and simply use the values from the grid cells that were showing subsidence. You could even determine the minimum and maximum subsidence per 100 x 100 m grid cell using the data from the 20 x 20 m cells. By taking a mean value, I wonder if you might be smoothing out results where the palsas are not filling the 100 m x 100 m grid cell. If I misunderstand your process, it needs to be more clearly described. When you resample, you create a new value. Perhaps you used the 20 x 20 m product for some processes, and 100 x 100 m in others? It needs to be clear what you used and when – it can have a large effect on your results.*

Response: Thank you for bringing this to our attention, there was an error in the methodology section, for the majority of the results we have shown did not resample the InSAR data to the 100 X 100m palsa raster cells. The results we have shown all clipped the InSAR dataset with the palsa raster as you have suggested. This was reported in the first line of section 2.3 of the methodology which has now been changed to reflect this. The only result that relied on resampled cells was figure 6 which used the resampling to compare the palsa statistics in these cells. We have changed the structure of the methodology to account for this with sub-sections for each form of analysis. In that case, it is true that we could have used the minimum and maximum values but because we are comparing against percentage palsa, the minimum or maximum values would not help us study how the amount of palsa impacts the overall ground motion for these cells.

*Comment 11: Line 203 says that the frequency numbers are derived from the 100 x 100 m resampled raster. But Fig 3 showing the frequency says it is based on the 20 x 20 m grid cells.*

Response: The full response to this mistake can be found in our reply to comment 10, however, Figure 3 was calculated using 20 x 20m grid cells and therefore the figure caption is correct.

*Comment 12: Line 233 – you say you used modelled permafrost probability distribution and make a reference to Obu 2018, however I didn't find it clear that you were using their data set. Please give a little more information – also, did you use the 1km grid cell data set and in that case what was your resampling method to 100 m cell size?*

Response: We agree that this was not as clear as it should have been. We have moved the reference up and rephrased the sentence on line 228 to state that we resampled all the data sources (Roughness, InSAR, and Permafrost Probability) to the 100m grid cells. Roughness and InSAR used mean values of the contained 2, and 20m cell size respectively. While Permafrost probability also used a mean value but for most cases due to the large cell size this only required a single probability value.

*Comment 13: Line 248 – Is this first number across all of the palsas, or all of the grid cells containing a certain percentage of palsas (in this case, what is the %?) or all of the grid cells within a certain area? Also, is this based on the 100 x 100 m grid cells that you have resampled? Because that might explain a lot when you are looking at areas with small isolated palsas, or even narrow ridge palsas.*

Response: In response to another review comment we have changed this text to read across all palsa cells measured in northern Sweden. We hope with the changes to the methodology it will be clear now that we have measured every palsa cell before we focused in on the eight sites. This is carried out at the 20 by 20 m grid cell level, although we accept this may still be at a larger scale than some of the scattered Palsa.

*Comment 14: Fig 6 – It seems the roughness index gave a higher value at the edge of the palsa, so if the roughness is indicating palsa edges only (and not the body of the palsa where there isn't a high roughness value), then you wouldn't expect to see the best correspondence with subsidence. It would be a bit noisy, as it is here. The result would be clearer if you had a clear geospatial layer delineating palsa from fen, perhaps by segmenting the palsas (based on roughness and elevation). Then this relationship would look better. That should be in the Discussion. I think the roughness index helps you identify some of the edges of palsas, but doesn't go as far as you need it to to give you a good map of palsa locations.*

Response: We agree with the reviewer's assessment here and we have mentioned the potential of further analysis using the roughness measure on line 457 in the future but have also added additional text to suggest what could be done to improve the correlation and use these as delineating layers in future research as has been suggested.

*Comment 15: Table 3 - Ok, so I have these data, and I looked up the Min daily temperature for Naimakka and found that it was -41.20 on 2021-12-06 (and the single Minimum temperature occurred on this day at -43.40). This doesn't match what you have in Table 3, which I am reading as being the lowest daily (averaged hourly) temperature that occurred between 01-01-2000 to 12-31-2021. Is there a misunderstanding of the terms or years given in the Table text? I did not check any other numbers.*

Response: The data we have were the minimum and maximum for each day, downloaded pre-calculated from SMHI. We exclusively looked at the max temperature per day because we wanted to show likelihood of melting permafrost. So, the minimum is the warmest temperature on what is effectively the coldest day. The table caption on line 358 reads "Maximum, minimum and the inter-quartile range [are] of daily maximum temperature" we have added the word "are" in to try and separate the daily maximum temperature out better and make it more readable.

*Comment 16: In the Discussion I think you should discuss the error levels of ASPIS InSAR as found by other studies. You weren't able to measure elevations and subsidence in this study to compare to what you found with InSAR, but other studies in other ecosystems have looked at the accuracy of InSAR and assessed the potential error. How reliable are your estimates, and how do you judge that? For example, Line 435 and 436 states "However, the high precision of the change in vertical position means that InSAR is an important tool to employ to detect the initial stages of large-scale permafrost degradation." – I think you need a reference here since you don't yourself do an accuracy assessment.*

Response: Thank you for this suggestion. Based on your feedback, we now include standard error rates so that an accuracy assessment of our results can be made. The vertical accuracy of the InSAR data is best represented using the standard error. The standard error for each palsa complex has now been included in a new column titled 'Mean standard error (mm yr$^{-1}$)' (line 272) and we will discuss the implication of this within the discussion. We conclude that because 69% of ground motion rates are within the margins of the mean standard error, there is greater confidence in the direction of surface motion as opposed to the magnitude. We hope the reviewer is satisfied with this interpretation.

*Comment 17: Line 424 – In this part of the Discussion, do you have any idea whether the dates of the SAR images in the stack can influence that there may be uplift? Or do all dates weigh in evenly in the InSAR stack? If you use images from April, May and June, and it has been a cold spring, for example?*

Response: All dates have equal weight in the InSAR stack and images are selected during snow-free periods. Whilst it is possible that anomalous climate conditions may have skewed results to show overall uplift, figure 8 shows variability but no clear trends in temperature, snow depth and precipitation from 2017-2021. Instead, we conclude the any uplift detected is a product of short-lived frost mounds that can form temporarily in the palsa system, or from rising water levels of flooded parts of the peatlands and accumulation of plant residues (Discussion paragraph 6, line 437). Since the palsa surface becomes more structurally and topographically complex with degradation, we expect that this uplift signal is a product of increased surface complexity with vegetation succession. We hope the reviewer is satisfied with our interpretation.

**Grammatical comments**

*Comment 18: Line 24 – something to fix with the grammar of the sentence*

Response: Thank you for this observation we have replaced this sentence with "We show that 55% of Sweden's eight largest palsa peatlands are currently subsiding, which can be attributed to these permafrost landforms and their degradation." Which will fix the grammar of this sentence on line 23.

*Comment 19: Line 30 – towards*

Response: We have made this change on line 31.

*Comment 20: Line 145 and 149 – consistency with the term plateaux or plateau.*

Response: We have made this change throughout.

*Comment 21: Line 179 – reference the SNAPHU algorithm (did you use the SNAP plugin?).*

Response: Thank you for pointing out the lack of reference, we have added the original SNAPHU algorithm reference (Shen et al., 2002) on line 178 and in responding to comment 6 we have explained how the algorithm was used in this context.

*Comment 22: Line 189 - It's a mouthful, but Lantmäteriet is officially called "The Swedish Mapping, Cadastral and Land Registration Authority" in English.*

Response: We have made the change to the longer official title to be more accurate.

*Comment 23: Line 198 – I would re-write as "… Kiruna, Karesuando, Saarikoski and Naimakka." Since the last two are two separate weather stations.*

Response: We have made this change on line 203.

*Comment 24: Line 237 – resampleto … resampled to*

Response: Thank you we have fixed this error on line 229.

*Comment 25: Line 248 – make sure the – sign before 9.9 stays with the number. It took me a couple of reads before I saw it.*

Response: This has been replaced with a non-breaking hyphen sign to stop this occurring.

*Comment 26: Line 249, 251 – palsa should be palsas (if plural)*

Response: The first of these has been changed to "palsa raster cells" with the plural attached to cells in response to an earlier comment. The change to plural "palsas" will be made for the line 268.

*Comment 27: Figure 2 – the figure text should mention that this is based on data for the years 2017-22. Also, the images are so small, it is hard to see the values of the grid cells. Can you make them a bit larger?*

Response: We have added the years over which InSAR surface motion was measured into the caption of figure 3 and will make this image a full page image to make it easier to interpret.

*Comment 28: Line 313 – "… subsidence proximal to …"*

Response: This grammatical error has been corrected by removing the unneeded word "in" on line 326.

*Comment 29: Line 334 – meteorological (spelling)*

Response: Thank you for noticing this, we have amended it on line 348.

*Comment 30: Fig 7 text - Is that daily snow depth measurements?*

Response: Yes these are all mean annual daily variables and the caption has been changed as such "Figure. 8: Mean annual a) daily maximum temperature, b) daily snow depth on the ground, and c) daily precipitation at the meteorological stations in the study region (SMHI 2022)." On line 358.

*Comment 31: Line 359 – "… the northern-most of the three weather stations with long-term records …"  Since Naimakka is further north, good to make the distinction that Karesuando is furthest north of the three….*

Response: We agree this would make our meaning easier to follow and have changed the sentence to "the northern most weather station of the three with long term records available" on line 369.

*Comment 32: Line 362 – add "for any of the sites" to make this clear. Also, what temperature? Mean annual temperature? Specify.*

Response: We have added "for any of these sites" to the beginning of the paragraph. We also specify that these are mean annual variables that did not show clear temporal trends on line 372.

*Comment 33: Line 370 – remove the comma "on-going subsidence"*

Response: Thank you for the correction we have removed this comma.

*Comment 34: Line 441 – something missing from a sentence.*

Response: The sentence has been changed to "However, our large-scale assessment of permafrost subsidence provides a baseline to direct, and compare, against future fieldwork monitoring in northern Sweden." to fix the error. This can now be found on line 456.

*Comment 35: Line 446 – Why Arctic DEM and not Copernicus DEM (see for example Karlson et al., 2021. https://www.diva-portal.org/smash/get/diva2:1617869/FULLTEXT01.pdf  Or Similar.*

Response: We would agree that the utilisation of more DEMs would provide better coverage but unfortunately, in the study referenced the authors compared these DEMs by aggregating the 2m Arctic DEM to 10m spatial resolution to compare to the Copernicus DEM. We would assume that these roughness trends needed to see the palsa would be too smooth over a 10m spatial resolution and therefore the Copernicus product could not be used here. It is also worth noting that the referenced article promotes the Arctic DEM for having a high vertical accuracy and therefore we have included this reference in the text on line 463.

*Comment 36: Line 495 – replace the ??*

Response: These will be replaced with the Sentinel 2 track numbers used in the project, which are 168 and 66 on line 516.

*Comment 37: Line 685 – terrain using*

Response: Thank you, a space has been added between these two words where it should be. Now line 715.

*Comment 38: Line 699 – thermal not theral*

Response: This correction has been made on line 729.